# Faecal Cortisol Metabolites as an Indicator of Adrenocortical Activity in Farmed Blue Foxes

**DOI:** 10.3390/ani11092631

**Published:** 2021-09-07

**Authors:** Eeva A. Ojala, Mika Kurkilahti, Anne Lene Hovland, Rupert Palme, Jaakko Mononen

**Affiliations:** 1Faculty of Veterinary Medicine, University of Helsinki, 00790 Helsinki, Finland; 2Natural Resources Institute Finland (Luke), Itäinen Pitkäkatu 4 A, 20520 Turku, Finland; mika.kurkilahti@luke.fi; 3Department of Animal and Aquacultural Sciences, Faculty of Biosciences, Norwegian University of Life Sciences, P.O. Box 5003, NO-1432 Ås, Norway; anlhovla@online.no; 4Department of Biomedical Sciences, University of Veterinary Medicine, Veterinärplatz 1, 1210 Vienna, Austria; Rupert.Palme@vetmeduni.ac.at; 5Natural Resources Institute Finland (Luke), Halolantie 31 A, 71750 Maaninka, Finland; jaakko.mononen@luke.fi

**Keywords:** faecal glucocorticoid metabolites, *Vulpes lagopus*

## Abstract

**Simple Summary:**

The measurement of faecal cortisol metabolites (FCMs) is increasingly used to monitor physiological stress responses in different animal species. Before this method is applied in coming stress- and welfare-related studies of farmed blue foxes (*Vulpes lagopus*), a species-specific validation is first required. In the current study, a 5α-pregnane-3ß,11ß,21-triol-20-one enzyme immunoassay was found suited to measure FCMs and thus hypothalamus–pituitary–adrenal (HPA) axis activity in farmed blue foxes. FCMs can therefore serve as a valid indicator of stress in future welfare studies of blue foxes.

**Abstract:**

Welfare studies of blue foxes would benefit from a measurement of faecal cortisol metabolites (FCMs) as a non-invasive, physiological stress parameter reflecting hypothalamus–pituitary–adrenal (HPA) axis activity. Before implementation, a species-specific validation of such a method is required. Therefore, we conducted a physiological validation of an enzyme immunoassay (EIA) to measure FCMs in blue foxes. Twenty individuals (nine males and eleven females) were injected with synthetic adrenocorticotrophic hormone (ACTH) and faecal samples were collected every third h for two days. The FCM baseline levels were assessed based on the first sampling day (control period, 144 samples), followed by the ACTH injection and the second day of sampling (treatment period, 122 samples). FCMs were analysed with a 5α-pregnane-3ß,11ß,21-triol-20-one EIA. We compared the estimated mean FCM concentrations of the treatment samples to the baseline average. All samples for the two periods were collected at the same time of the day, which enabled to test the data also with an hourly pairwise comparison. With the two statistical approaches, we tested whether a possible diurnal fluctuation in the FCM concentrations affected the interpretation of the results. Compared to the baseline levels, both approaches showed 2.4–3.2 times higher concentrations on time points sampled 8–14 h after the ACTH injection (*p* < 0.05). The estimated FCM concentrations also fluctuated slightly within the control period (*p* < 0.01). Inter-individual variations in FCM levels were marked, which highlights the importance of having a sufficient number of animals in experiments utilising FCMs. The sampling intervals of 3 h enabled forming of informative FCM curves. Taken together, this study proves that FCM analysis with a 5α-pregnane-3ß,11ß,21-triol-20-one EIA is a valid measurement of adrenocortical activity in the farmed blue foxes. Therefore, it can be utilised as a non-invasive stress indicator in future animal welfare studies of the species.

## 1. Introduction

The blue fox, the blue colour morph of the Arctic fox *(Vulpes lagopus)* [1], is the most important farmed fox species in the world, with an annual production of more than 10 million pelts (in 2019) [2]. Foxes have been farmed for one century [3], and the relatively short domestication history, together with traditional housing in cages and management procedures, have raised concern over the welfare of farmed foxes [4]. Thus, alternative methods to strengthen the assessment of fox welfare are warranted.

The welfare research of blue foxes has covered topics such as housing environment [5,6,7], temperament [8], human–animal relationship [9], handling [10], and social housing [11,12]. The physiological welfare measurements have relied strongly on the functioning of the hypothalamus–pituitary–adrenal axis (HPA; see the references above for examples). The main glucocorticoid secreted by the HPA axis in almost all mammal species, also in blue foxes, is cortisol [13]. In the blood, baseline cortisol levels have been measured [5,8,9,10] and change after ACTH challenge or an acute stressor [7,9,11,12]. Furthermore, postmortem adrenal mass has been used to reflect adrenocortical activity when alive [5,7,8,9,11,12]. Blood cortisol has dominated, although the invasiveness of this tool has been acknowledged already for some time. Moreover, the stress experienced by repeated blood sampling may in many cases raise questions about its validity as a welfare indicator [13,14,15].

In recent decades, sample materials other than blood for analysing cortisol (or its metabolites), such as faeces, urine, saliva, or hair, have gained popularity in animal welfare studies [13,14]. A benefit of all these tools is that they integrate cortisol secretion over longer periods. This smooths the effects of the originally episodic secretion of plasma cortisol [16] and reduces variation in the cortisol data [13,14]. From these complementary sampling materials, the least invasive are collecting urine and faeces since animal handling is not necessary or is minimal. For farmed foxes, urinary cortisol has been utilised as a measure of stress to some extent [5,6,7,9,11,12], but the sampling of urine requires special equipment, such as trays, bottles, and gauzes [6]. Instead, faeces can be easily collected from cloths mounted under the cages.

The measurement of faecal cortisol metabolites (FCMs) requires species-specific validation [13]. It has already been validated as a stress measurement in two farmed fur animal species, the silver fox [17] and the mink [18]. FCMs have also been measured in blue foxes [19], but without any validation of the assay. A recent FCM validation study used the farmed white colour morph of *V. lagopus*, the “white polar”, as a model animal for the wild Arctic fox [20]. In the wild, the white colour morph comprises over 99% of the populations in the continental ranges [21], while the blue foxes mainly occur in coastal habitats [1]. Compared to the farmed blue fox, the white polar is a rarely farmed type and it has an appearance and size closer to the wild Arctic fox. The blue foxes have been successfully bred, e.g., for larger size, which has brought some issues typical for large size, such as leg weakness [22]. Due to the different origins and breeding schemes, there is uncertainty about whether the HPA axis functions similarly in the two morphs.

The adrenal gland has an essential role in the control of stress and immune responses of the body [23]. Of the gland, 90% is composed of the adrenal cortex that is specialised to produce mineralocorticoids, adrenal androgens, and glucocorticoids (GCs), such as cortisol. The synthesis of cortisol is controlled by the pituitary hormone, adrenocorticotrophic hormone (ACTH) [14]. The secretion of ACTH depends under normal circumstances on two neuropeptides, corticotrophin-releasing hormone (CRH) and vasopressin, synthesised by the paraventricular nucleus of the hypothalamus (PVN). The PVN converges signals from numerous inputs and, consequently, the HPA axis is sensitive to various stimuli from external and internal origins. GCs, including cortisol, are metabolised by the liver and the metabolites are excreted for the most part via the bile. After a species-specific time delay, an increased level of FCMs can be detected in the faeces [13]. The most widely used experiment for physiological validation to determine GCs is an ACTH challenge test, where the HPA axis is pharmacologically stimulated [13,14,24]. In the test, the animals are injected with ACTH, causing the adrenal cortex to release GCs into circulation. The peak increase in FCM levels caused by the ACTH injection demonstrates that the FCMs studied are valid for measuring FCM levels, and thus HPA axis activity [13]. Cortisol metabolites are commonly measured with enzyme immunoassays (EIAs), where group-specific antibodies have been found most useful [13,25]. Among those, a 5α-pregnane-3ß,11ß,21-triol-20-one EIA, which was first described by Touma et al. [26], has recently been validated in *V. vulpes* [17] and the white colour morph of *V. lagopus* [20]. Thus, it is a good candidate for analysing adrenocortical activity in blue foxes.

Monitoring FCM concentrations could have potential as a valuable tool in stress- and welfare-related studies of farmed blue foxes. Our study aimed to conduct a physiological validation of the 5α-pregnane-3ß,11ß,21-triol-20-one EIA to measure FCMs as an indicator of HPA axis activity in farmed blue foxes. We also aimed to examine issues related to the possible diurnal pattern of cortisol secretion in blue foxes and its implications to the statistical approaches, as well as optimising faecal sampling intervals.

## 2. Materials and Methods

### 2.1. Ethical Statement

The procedures used in the present study were approved by the Animal Experimental Board (ELLA) in Finland (ID 11562), in accordance with the guidelines provided by the EU Directive 2010/63/EU for animal experiments.

### 2.2. Animals and Housing

The validation experiment was carried out on Kannus Research Farm Luova Ltd., Finland with nine male and eleven female blue foxes that had been born and kept in traditional fox farming conditions. The experimental animals were 8–9 months old at the time of the experiment, i.e., ‘sub-adults’ a few months before their first breeding season. The animals were housed singly in wire mesh cages with plastic-coated floors (85–100 cm above the ground) and the cages were situated next to each other on the left side of a two-row shed, i.e., a light building with a roof but no walls. The males were housed in cages 1 to 9, the females in cages 11 to 21, and the remaining cages of the same shed (cages 23 to 52 in the left row and cages 53 to 104 in the right cage row) were housed with other young breeding females. The cage area per animal was 1.2 m^2^ (width: 107 cm × length: 114 cm) and the height of the cages was 72 cm. Each cage included a bovine femur bone as an activity object and an elevated resting platform (made of plastic-coated wire mesh, length 107 cm, width 28 cm) mounted on the left sidewall of the cage 46 cm above the cage floor. The animals were fed once a day between 10.00 a.m. and 11.00. a.m. The feed was commercial, fresh food paste (DM 33,1%, 3880 kCal/kg/DM), produced by a Finnish feed kitchen specified for fur animal feed. Of the metabolisable energy, 44.5%, 34.7%, and 20.8% was from protein, fat, and carbohydrates, respectively. The main ingredients of the feed were slaughterhouse offal, fish, fish offal, and cereals. The animals had ad libitum access to drinking water, provided from an automated non-freezing drinking system with nipples.

### 2.3. Study Procedure and Collection of Faecal Samples

#### 2.3.1. Experimental Design

The experimental animals were moved into the experimental cages in mid-December 2018, 53 days before the start of the experiment, except for two females that were moved 14 days before the start. During this habituation time, the daily management routines were standardised to correspond to the routines during the experimental period to minimise the effects of disturbances as confounding factors. In the week before the experiment, the feed portions of the foxes were raised gradually to increase the probability of a greater number of faecal samples, but the portion size was kept reasonable considering the production cycle and the time of the year. The increase was from 350 to 450 g and from 400 to 500 g for the females and males, respectively. The final portion size was reached two days before the start of the experiment and kept constant throughout the study (no leftovers).

All animals from the experimental shed were habituated to the extra activity in the shed by three in-cage behavioural tests. In addition, the experimental animals were habituated to the sampling procedures by health checks and experimental preparations. The behavioural tests were conducted 13 days (Titbit test, TBT [15]) and 12 days (Stick test, ST [27] and Subjective evaluation of human–animal relationship, SEH [28]) before the start of the experiment. All behavioural tests included human–animal contact, which lasted 10–30 s per animal, depending on the test. In addition, the animals were rewarded in the TBT with a small dog biscuit. The behavioural tests were a part of a different study where all the animals on the farm were tested with the same three tests. The health checks and the experimental preparations were carried out on 15, 8, 7, 2, and 1 day(s) before the start of the experiment. The first two health checks were performed from inside of the shed, the third health check from both inside and outside of the shed. In the third check, the inspector also went under the cages to habituate the foxes to the sampling procedure. Two days before the start of the experiment, white filter cloths were placed on the ground under the cages to adapt the foxes to them, and the next day the cloths were removed. The animals were not handled in any other way during the habituation period.

#### 2.3.2. Collection of Faeces

The filter cloths were placed under the cages on 13 February 2019 at 5 am, i.e., 3 h before the first faecal sample point. We used the same cloths throughout the experiment. We collected the first faecal samples at 8.00 a.m. and subsequently every 3 h for 24 h to evaluate baseline FCM levels (control period). The decision of the 3 h time points was based on the suggestion made by Hovland et al. [17] for the silver fox. The samples were collected into plastic bags with plastic spoons (1 spoon per animal, the spoons were cleaned between the time points) and put into a freezer (−20 °C) within a maximum of 30 min after collecting the samples. The control period was followed by the ACTH injection and a similar sampling procedure for another 24 h (treatment period). We collected the samples for control and treatment periods at the same h. The time points were named as a continuum from the start of the experiment (control period: 0, 3, 6, 9, 12, 15, 18, 21, 24; treatment period: 27, 30, 33, 36, 39, 42, 45, 48) where the number 0 refers to the first sampling point. The ambient temperature was registered at every sampling point, and it varied between −10 and 3.5 °C during the two experimental days, being −1 °C on average. Due to a low temperature, the risk for bacterial degradation of the samples was low.

#### 2.3.3. ACTH Injection

The ACTH injections were started at 9 a.m. on the 14 February and conducted by a group of three persons. The experimental foxes were treated in increasing cage number order. The animal keeper caught the animal with a traditional neck tong, took it out of the cage, and restrained it while the veterinarian injected (2 mL syringe, 16 mm needle) 1 mL Synacthen^®^ (0.25 mg mL^−1^ tetracosactid; Alfasigma, Bologna, Italy) intramuscularly in the upper thigh (biceps femoris). The third person recorded the duration of the treatment. The treatment lasted approximately 30 s per fox; thus, all animals were treated within 15 min.

### 2.4. High-Performance Liquid Chromatography (HPLC)

To characterise the measured metabolites, we performed a reverse-phase high-performance liquid chromatographic (RP-HPLC) separation (similar as for the Arctic fox [20], for details see [26]) of a peak concentration sample of a male and female blue fox.

### 2.5. Analysis of Faecal Cortisol Metabolites

Faecal samples were transported on dry ice from the experimental farm to a Finnish laboratory (Movet Ltd., Kuopio, Finland) for extraction in February 2019. The samples were stored at −20 °C in the laboratory until the pre-treatment in March. The frozen samples were thawed in a fume cupboard at 60 °C for approx. 45 min. The thawed samples were homogenised inside plastic bags and an amount of 0.5 g per sample was extracted with 5 mL of 80% methanol by shaking with a hand vortex mixer for 1.5–2 min before centrifugation at 2500× *g* for 15 min (described by Palme et al. [13,29]). After centrifugation, an amount of 0.5 mL per supernatant sample was transferred into Eppendorf tubes. The samples were evaporated in heating blocks until complete dryness (4 to 6 h). Then, the dried extracts were sent to the Vetmeduni Vienna for further EIA analysis.

The dried down supernatants were dissolved again in methanol (0.5 mL 80%) and then diluted (1:20) with assay buffer. The supernatants were analysed with a 5α-pregnane-3ß,11ß,21-triol-20-one EIA, first described by Touma et al. [26]. The selection of the metabolite was based on the results by Hovland et al. [17], where the 5α-pregnane-3ß,11ß,21-triol-20-one EIA proved well suited for silver foxes. All analyses were performed blindly, i.e., no information was known about the identity of the samples. Intra- and interassay coefficients of variation of a high- and low-concentration pool sample were <12% and <15%, respectively. The detection limit of the FCM analysis was 2.2 ng/g. Serial dilutions of two high-concentration extracts were parallel to the standard curve. Detailed cross-reactions of the antibody with several glucocorticoid metabolites can be found in Touma et al. [26], and also in Santamaria et al. [30].

### 2.6. Statistical Analyses

We used SAS/STAT software (version 9.4, SAS Institute, 2018) in the statistical processing of the data.

The distribution of FCM was positively skewed and a log-transformation was applied to the data. A general linear mixed model was fitted using a normal distribution (identity link) [31]. A Kenward–Rogers approximation was used for estimating degrees of freedom. The model fit was checked from the shape of Pearson residuals and the observed vs. predicted plots. The modelling was performed by the GLIMMIX procedure.

The study design was a two-period repeated measurement structure where the period (1st control, 2nd treatment) was nested with the time points.

The structure of the model was:
log (FCM) = intercept + sex (F) + period (time point) (F) + time point (Rperiod)
where F stands for a fixed effect and R for an R-side repeated effect over the time points. Fixed effect period was nested with time points and the repeated time point effect was grouped by the period. The covariance structure was modelled as grouped by treatment period 1st order autoregressive variance–covariance matrix structure (ARH1), allowing different variances for both periods and each time point (Appendix A). The animal ID was the subject [31], which allowed an individual specific random effect with autocorrelation over time described above. The chosen covariance structure and Kenward–Rogers correction together adjusted the degrees of freedom (standard errors, confidence intervals etc., Appendix A) for each statistical test with the number of observations (Appendix A) per test at each time point and variance modelled for that time point (Appendix A).

Control and treatment periods had a different covariance structure (time point (Rperiod)) and this model allowed estimation of variation in individual levels for each time point in both periods. Statistical comparisons were based on two predefined approaches:Baseline average. Mean value of the control period (a baseline) vs. mean value at each time point (27–48) separately under the treatment period (*t*-tests by contrasts)Hourly pairwise comparison. Pairwise comparison (eight *t*-tests by contrasts) of the time points between the same time of day from the two periods.

Baseline average compares the one mean value of all time points within the control period to each mean value of the time points under the treatment period. This is different compared to the approach which is often used in other FCM-related studies [13], i.e., no iterative procedure to exclude outliers was performed, but the whole dataset was analysed. Detection of possible outliers was performed with residual analysis after fitting the model (see below). The baseline average approach has an assumption that there is no, or minimal, diurnal fluctuation in FCM values. Hourly pairwise comparison compares the mean values of the control and treatment periods under the same time points, noticing the effects of any diurnal rhythm. Additionally, post hoc comparisons (36 *t*-tests) within baseline averages (control period) were conducted. Due to multiple *t*-tests, we adjusted *p*-values and confidence intervals in each test-set by using a simulation-based correction in a step-down fashion ([32], SAS 9.4 documentation: PROC GLIMMIX, ESTIMATE-statement, ADJDFE = row, ADJUST = sim, STEPDOWN(ORDER = *p*-value)). Conclusions from statistical comparisons are based on adjusted *p*-values and confidence intervals.

Statistical estimation and hypothesis testing were performed on log-transformed data and the estimated means, the differences, and the endpoints of the confidence intervals were converted to the original scale using the exponential function. Log-scale difference zero converts to a data-scale ratio of one. If one is not included in the 95% confidence interval of the ratio then *p* < 0.05. The ratio gives a multiplier (with a confidence interval) to compare two treatments: how much higher (or lower) the treatment of interest is.

During the analysis process, a sex × period (time point)-interaction effect was analysed but left out based on AICc criterion. Additionally, a model without sex was analysed but the effect was left in the model based on AICc criterion. The residual distribution was symmetrical, where the observed vs. predicted values plot was distributed along the diagonal and the model fit was good in individual log-scale subject-specific profile figures. The effect of outliers was checked by reanalysing the dataset in which observations with Pearson residual > |2.5| were excluded. This condition covers about 99.9% of normally distributed values. The outcomes of the two analyses differed slightly within baseline post hoc analysis (Appendix A) and therefore the original dataset was used for all analysis.

## 3. Results

### 3.1. ACTH Challenge Test

FCM concentrations remained above the baseline throughout the treatment period (Figure 1). Both statistical approaches showed the highest concentrations 5–14 h (h 14:00–23:00) after treatment, where the four time points had 2.3–3.2 times higher values than in the control period (Figure 2). The most marked differences were detected between 8 and 14 h (h 17:00, 20:00, 23:00) after ACTH injection (P(baseline comparison) = 0.014, <0.001, 0.001 Appendix A; P(hourly pairwise comparison) = 0.043, 0.001, 0.001, Appendix A), including the FCM peak concentration, with a ratio 3.0–3.2, which was reached 11 h after the injection (Figure 2: h 20:00, Appendix A). There was a slight discrepancy between the baseline average comparison and the hourly pairwise comparison results, the latter showing a 3 h delay in the FCM response as compared to the former (Figure 2). This is probably explained by the slight diurnal variation within the control period (Figure 1; Appendix A). The FCM concentrations were highest at 14:00 and the peak was 1.9 times higher than the lowest value measured 12 h later at 02:00 (*p* = 0.010). However, most of the comparisons between the time points within the control period did not differ from each other and only the most marked differences are shown in the Appendix A. Sex had no clear impact on FCM concentrations (*p* = 0.083, Appendix A).

### 3.2. Individual Variation

Inter-individual differences in FCM concentration were marked, especially after the ACTH injection (Figure 3). Some animals had only a modest FCM increase after the ACTH administration, and animal 13 seemed to have no increase at all. Animal 3 had an unexpected increase on the control day. Nevertheless, all animals were included in all statistical analyses. The exclusion of the two deviating animals would not have had an impact on the main results of our study (Appendix A).

### 3.3. High-Performance Liquid Chromatography (HPLC)

HPLC immunograms revealed that several metabolites were picked up by the EIAs (Figure 4) in both sexes. The utilised 5α-pregnane-3ß,11ß,21-triol-20-one EIA showed higher immunoreactivity compared to the cortisol EIA. Due to their elution position, the main metabolites were unconjugated steroids, but unmetabolised cortisol was absent (or at too low a concentration to be measured).

### 3.4. Faecal Sampling Interval and Success

The 3 h sampling interval led to a situation where 80% (on average 16 of 20 sampling attempts per sampling point) of all samplings were successful, and 20% turned out to be ‘unnecessary’ due to lack of faeces. In the control period, this percentage (so called ‘hits per cent’) was 82% (average 16.3) and 78% in the treatment period (average 15.6). In some cases, the faeces were a bit loose but clear signs of diarrhoea or other diseases were not observed.

## 4. Discussion

Our study aimed to evaluate whether FCMs could be a valid tool for estimating adrenocortical activity in blue foxes, with an overall purpose to use FCMs as a complementary indicator of stress in future welfare studies. We observed an expected increase in the FCMs after the ACTH treatment. Thus, we demonstrated that the utilised 5α-pregnane-3ß,11ß,21-triol-20-one EIA can be used as a non-invasive method to monitor adrenocortical activity in welfare studies of blue foxes in the future.

Every assay requires species-specific physiological and/or biological validation [13]. Both the biological and the physiological validations of 5α-pregnane-3ß,11ß,21-triol-20-one EIA have already been conducted with *V. vulpes* [17] and the white colour morph of *V.lagopus* [20]. In both studies, the physiological validation was carried out by ACTH challenge tests, and the biological validation by using handling as a stressful factor. We minimised the number of stressful handlings and did not conduct biological validation or sham injections to study the effects of handling on FCM levels. Our physiological validation was sufficient to prove that the same EIA method also works in blue foxes. Similar FCMs are excreted in both morphs, shown by the HPCL immunograms [20]. The blue fox serves as a model for a standard blue fox in farm conditions, whereas the white morph “white polar”, serves as a model for wild Arctic foxes.

We tested whether the possible diurnal pattern of cortisol secretion might affect the interpretation of the results using two different statistical approaches. The comparison of the FCM levels after the ACTH injection pairwise to the control period takes into account the possible diurnal rhythm of cortisol secretion in 3 h intervals, whereas the comparison to the baseline average does not. The main results were not affected by the statistical approach. This might have been somewhat predictable since the time of the day does not have a strong effect on the excreted FCMs in blue foxes (Figure 1, control period). In contrast, in rats [34] and mice [26], the diurnal variation in FCMs is considerable, and hourly pairwise comparison could be more appropriate than comparison to the baseline average. Indeed, even our blue fox data showed that hourly pairwise comparison was more accurate, indicated by the narrower confidence intervals for the former.

FCM studies will probably include individuals with unexpected responses, and a crucial question is whether these individuals should be omitted (as outliers) from the final statistical analyses or not. In our study, two animals did not show the expected pattern in FCM concentrations. Furthermore, some individuals showed a two-humped increase in the treatment period. The underlying reasons for the unexpected responses are merely up to speculation and could relate to an external stress-inducing factor in the control period (animal 3, Figure 3.), possibly a failed ACTH injection (animal 13), or an enterohepatic recirculation (two-humped increase, animal 5). From the welfare point of view, the high and low values may indicate impaired welfare [35]; the HPA axis of chronically stressed animals may become chronically elevated or the functioning of the entire HPA axis may diminish and/or have a flat diurnal pattern [36]. Therefore, some welfare studies may focus on individual glucocorticoid responses [35]. Sex can also affect glucocorticoid metabolism. The FCMs measured with the same assay with mice were affected by sex [26]. Our results did not reveal differences between sexes in FCM concentrations following ACTH injection, and neither did the study with silver foxes [17]. The time to reach peak FCM values after ACTH administration tended to be shorter for silver fox males. However, ‘the time to reach peak FCM values’ is rather based on an evaluation, because the animals do not defecate at predetermined times and intervals. Due to uneven sampling and unknown time of defecation within the sampling interval, we decided not to include this parameter in our results. In general, the sex differences may be somewhat masked by marked individual variation. All things considered, future welfare studies of blue foxes will benefit from presenting all individual FCM curves.

A previous FCM study with silver foxes [17] suggested that the sampling interval of 3 h could suffice for detecting the peak concentrations but, at the same time, avoids sampling in vain, i.e., when there are no faeces to be collected. We confirmed that this sampling interval with controlled feed portions of 450 g (females) and 500 g (males) worked well in blue foxes. All samples were successfully collected, despite the slight variation in the sample looseness. Due to the cold temperature (−1 °C on average) and the instant storing of the samples in the freezer, bacterial metabolism and its effects on FCM concentrations were avoided [13,29]. The highest concentrations were presumably detected, and the interval was sufficient to determine the individual FCM curves. The hits per cent of 80% in our study paralleled the results by Hovland et al. [17]. In their study, the hits per cent was 57–61% for the samples collected with intervals of 2 h after the ACTH injection and 82–91% for the samples collected with intervals of 4 h in the control period. Frequent sampling allows monitoring of both short-term and long-term endocrine changes [26]. As a non-invasive method, the repeated sampling of faeces is probably less stress-inducing than repeated blood sampling, where the method itself has been shown to cause stress in farmed foxes [15], raising questions about the validity of blood cortisol as a stress parameter [13,14]. Furthermore, the measurement of FCMs complies with the principles of the 3Rs [37], while it is non-invasive and a separate control group is not needed when the animal serves as its control.

Radiometabolism studies with the genus *Vulpes* would help to obtain more exact information about the time delay of FCM excretion. In these studies, the animals are injected with a radiolabelled steroid hormone that can be used to measure the time lag between the secretion of GCs into the blood and the excretion of their metabolites [13]. This species-specific time delay is comparable with gut passage time. The improved knowledge of FCM excretion time would also improve our interpretation of the relationship between HPA activity and behavioural activity. Elevated HPA axis activity may be associated with increased behavioural activity [38]. The behavioural activity rhythm of farmed blue foxes may be related to daylight and working hours on the farm [9,39]. Farm foxes typically have a regular feeding time, which has been shown to impact the release of cortisol in other farm animals [14]. Studies of behavioural activity rhythm in relation to time delay of FCM excretion could be advantageous in future welfare studies of *V. lagopus*.

## 5. Conclusions

With this study, we demonstrated that a 5α-pregnane-3ß,11ß,21-triol-20-one EIA proved well suited to determine FCMs and thus adrenocortical activity in farmed blue foxes. Accordingly, measured FCMs can be used as an indicator of stress in welfare studies of blue foxes.

## Figures and Tables

**Figure 1 animals-11-02631-f001:**
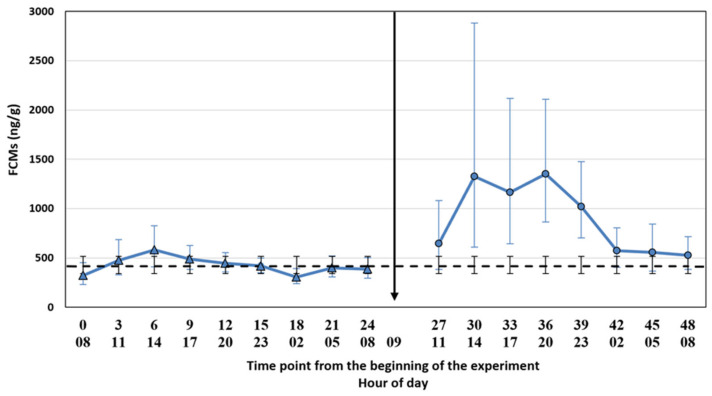
FCM mean concentrations with a 95% CI at each time point from 0 to 48 (blue line) and the baseline average (black dash line) with a 95% CI, both based on a model estimation on a data scale. ACTH was injected between 9:00 and 9:15 (arrow). Control period = time points 0–24 from the beginning of the experiment; Treatment period = time points 27–48 from the beginning of the experiment.

**Figure 2 animals-11-02631-f002:**
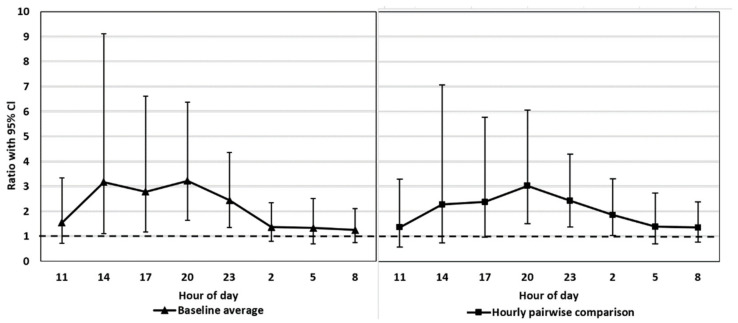
FCM concentrations based on model estimations of the two statistical approaches: 1. Baseline average comparison on the left (Appendix A) and 2. Hourly pairwise comparison on the right (Appendix A). Y-axis ratio: mean concentration(s) of control period (dashed line)/mean concentration at each time point of treatment period (95% Cl). The ratio signifies the estimated increase in the treatment period compared to the control period. For example, the peak FCM concentration at 20:00 was 3.2 (baseline average) or 3.0 (hourly pairwise comparison) times higher compared to the reference value from the control period. *p* < 0.05 if the lower endpoint of the confidence interval stays above 1, e.g., at 20:00 with both approaches.

**Figure 3 animals-11-02631-f003:**
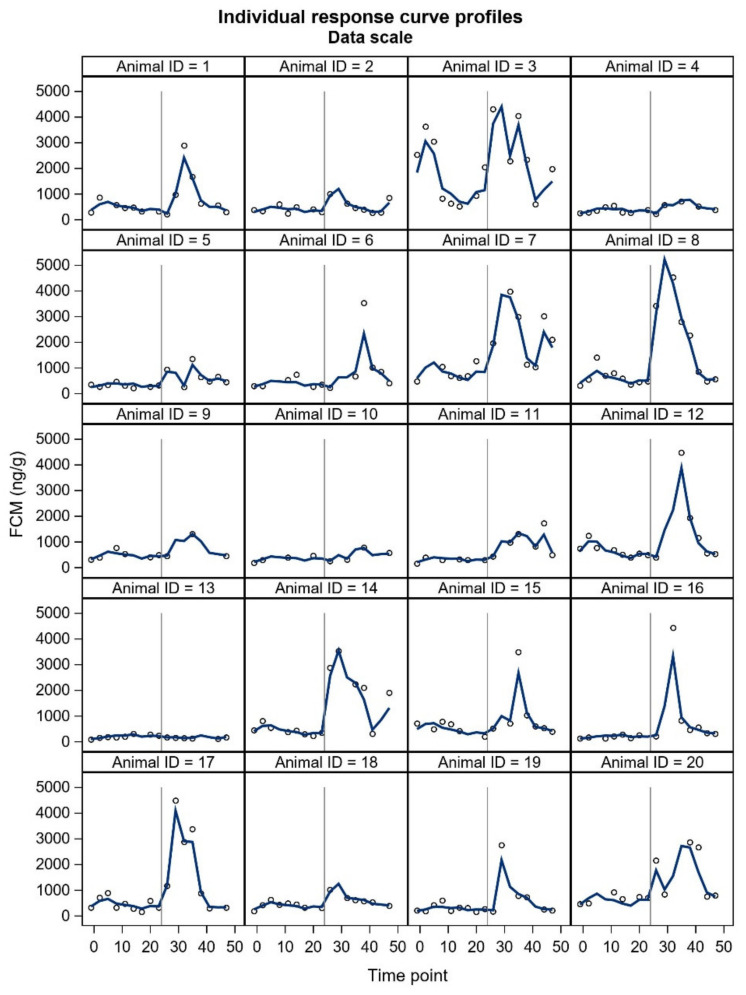
Individual curves of FCM concentrations based on the model estimation. The observations are presented with an open circle. The animal IDs from 1 to 9 represent males and the animal IDs from 10 to 20 represent females. The vertical grey line at time point 25 indicates the time of the ACTH injection.

**Figure 4 animals-11-02631-f004:**
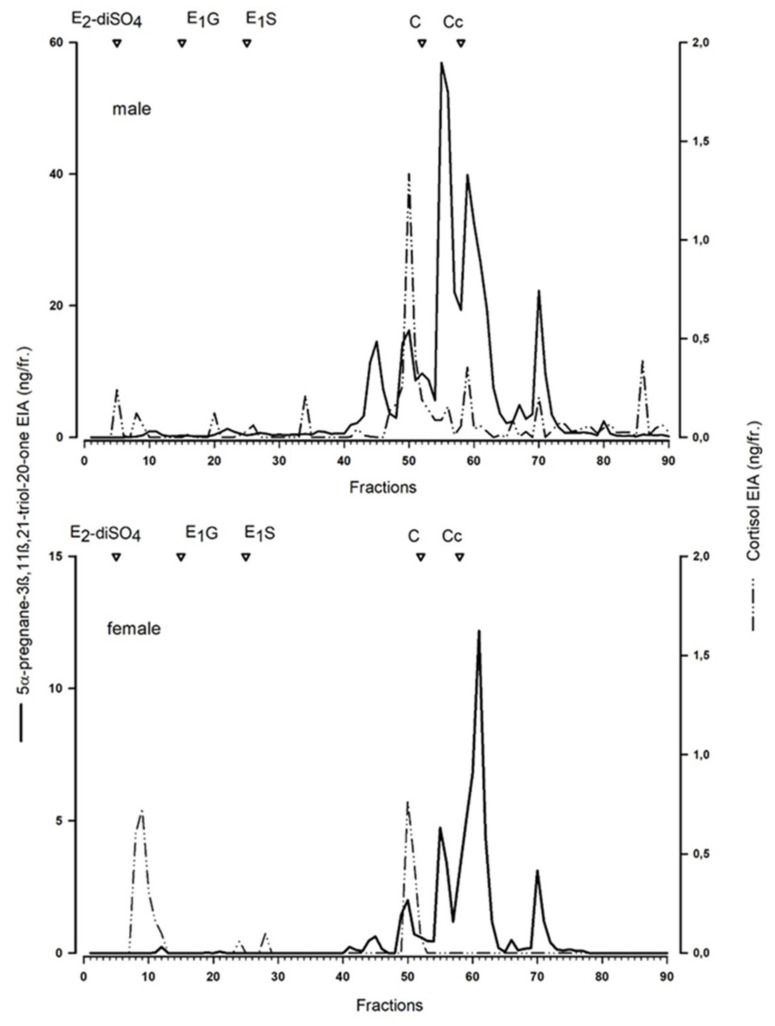
Reverse-phase high-performance liquid chromatographic (RP-HPLC) separation of faecal cortisol metabolites of a male and a female blue fox as measured in the 5α-pregnane-3ß,11ß,21-triol-20-one EIA (――) and a cortisol EIA (−··−). For details of both EIAs, see [26,33]. Open triangles mark the approximate elution positions of respective standards (E_2_ß-diSO4: 17ß-estradiol-disulfate, E_1_G: oestrone-glucuronide, E_1_S: oestrone-sulfate, C: cortisol, Cc: corticosterone).

## Data Availability

The data presented in this study are available in Suplementary Data (Appendix A).

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
