# Peer review of "Faecal Cortisol Metabolites as an Indicator of Adrenocortical Activity in Farmed Blue Foxes"

_animals, 2021, doi:10.3390/ani11092631_

Round 1

Reviewer 1 Report

This study explores the validation of a method of measuring fecal cortisol metabolites (FCM) in a blue color morph of farmed foxes (Vulpes lagopus), using an EIA which measures 5α-pregnane-3ß,11ß,21-triol-20-one EIA. While studies which appropriately validate the use of any measure of stress hormones in a species-specific manner are always appreciated, I am confused by the citation of a study in the same species which supposedly covers basically the same experiment and information, but in the white morph rather than the blue (reference 21). The study in white foxes has not yet been published (it’s listed as being under review), and there is overlap in the authors involved in both the white and blue fox studies, so it would probably produce a better paper if all the data from both of these color morphs were published together in the same paper, since they are the same species. It’s also very odd to cite a paper that is under review as if it has already been published; it should instead be cited as unpublished data.

Overall the data presented here is a valuable contribution to the literature on non-invasive analysis of glucocorticoid stress hormones in farmed animals, but would benefit from being combined with the data on white foxes. It would also benefit from a comparison to more standard measures of cortisol, such as plasma concentrations to check for correlation, if at all possible.

Abstract

  1. The abbreviations in the abstract should be defined
  2. Line 35: Could you clarify what you mean by “proper planning phase”?

Introduction

  1. In general, the Introduction could use some editing, it seems to meander and it’s not always easily apparent what the main point of each paragraph is. There are occasional issues with article usage as well.
  2. Lines 46-47: I don’t think you need this information here, it detracts from the main point of this paragraph. I suggest it be deleted or moved elsewhere.
  3. Lines 48-49: Maybe move this up, it’s the most important point in this paragraph
  4. Line 64: This seems like a stretch in reference to the most common media for measuring glucocorticoid hormones. There are many reasons to question the use of glucocorticoids as a measure of animal welfare, but the ability to get baseline plasma CORT isn’t really one of them. It’s well-documented in most mammals that if you obtain blood quickly (within a few minutes of disturbance) the plasma CORT concentration is at or close to baseline. Not that the stress of bleeding an animal is irrelevant; it’s a valid welfare concern in and of itself, but it’s unlikely to harm your CORT readings.
  5. Lines 80-82: I would rephrase this to emphasize how the blue foxes differ from the white, rather than how the other study’s animals differ from yours.
  6. Lines 83-84: The ACTH stim test, while widely used in research and veterinary medicine, is also not really “pharmacologically standardized” – it’s known to vary widely in individual response, and between seasons.
  7. Line 86: It would be a good idea to give a brief overview of the HPA axis first to better explain the ACTH stim test. Currently you introduce the ACTH stim test and then sort of work backwards to explain the HPA axis in a piecemeal fashion (for example, the adrenal glands are not mentioned), but it would make more sense if you explain the HPA axis first in a more organized, step-wise manner and then after that highlight the parts of the system you are working with in this experiment.

Methods

  1. Line 108: Change “was” to “were”
  2. Line 125: Could you provide more information about the commercial food paste?
  3. Line 141: Could you provide citations for the behavioral tests you used (TBT, ST, SEH)? Did these tests serve any other purpose aside from habituating the animals to human interaction?

Results

  1. 4: Please clarify in the figure legend which EIA is illustrated with a solid line and which with a dashed line. From the text I think the solid line is the 5α-pregnane-3ß,11ß,21-triol-20-one EIA, and the dashed is the cortisol EIA, but it’s not specified in the legend.

Discussion

  1. Lines 312-314: I don’t think you need to restate your methods here, I’d focus on summarizing how the results fit into your predictions.
  2. Line 319: “proved” is a bit strong to use here
  3. Lines 325-326: I’d delete this line, you’ve said it before and it’s fairly obvious. It might also be a good idea to merge this paragraph with the one before, I think they’re basically covering the same concept, that this method was previously validated in white morphs, but the data presented here indicates it also works in the blue morphs.
  4. Related to the above point – is there any reason to think it would work differently in the blue morphs? Is there anything special about them that might have changed the assay results at all? If so, I’d add that to the Introduction.
  5. Line 339: Please provide a citation for this here, or clarify if you’re referring to the results from the control period in this study.
  6. Line 345: Delete “The data of”
  7. Line 355: Replace “effect” with “affect”
  8. Lines 355-357: Did you consider controlling for food volume? Differences in food consumption can alter FCM concentrations, and might be masking sex effects.
  9. Lines 375-376: Ah, you are concerned about the usefulness of repeated bleeds in assessing CORT since each bleed would cause a stress response. That makes more sense. This should be clarified in the Introduction.

Reviewer 2 Report

This study aimed to validate an enzyme immunoassay to measure fecal cortisol metabolites in feces from blue foxes by performing a physiological validation.  An ACTH challenge was administered, and rigorous sampling procedures were used to observe the peak FCM concentrations resulting from the challenge.  Having completed this physiological challenge, the authors suggest that the measurement of FCMs in this manner can be used in future studies that focus on the welfare of farmed foxes.

This study is well designed and was performed on a reasonable number of animals.  The frequent sampling interval, particularly for fecal sampling, is commendable and allowed for the clear detection of FCM peaks.  The figures help to clearly display the results and provide excellent detail.  The manuscript would be improved by addressing some unclear aspects in the analysis and reporting of the data.

Major comments:

In describing the statistical approaches used to compare baseline to treatment conditions, the authors explain that the baseline average approach they use is similar to the approach commonly used in other FCM related studies (line 225), but the cited reference explains this approach for determining the baseline concentration, not comparing the baseline concentration to a treatment concentration.  Therefore, providing a citation where this mixed modelling approach is used in this context would be helpful.  Given the uneven sampling within individuals (not all sampling events yielded samples for analysis) and repeated measures, it is important to explain how these t-test based approaches would be robust to the limitations of the data, which would seemingly lead to higher probabilities of a type 1 error, especially with the number of comparisons that were made.  The method used to account for the repeated measures in the model could also be explained more clearly--was individual ID included in the model as a random effect?  

The inclusion of HPLC data strengthens the conclusions drawn, but the methods are only vaguely described, and there is little interpretation of the results.  The authors refer the reader to "details as reported in {24}" (line 184)--does this mean the same methods were used?  This could be more clearly explained.

There is no reported analytical validation of the assay.  A parallelism test would improve confidence in the results.  It is not clear if this test was performed using feces from the same species in the cited paper that is currently under review.  This should be addressed more clearly.

There are no reported intra- or inter-assay variations.  These must be reported to provide context for the change in FCM concentrations observed between conditions.

Ideally, the effect of handling for injections (sham injection) would have been compared to the baseline and the ACTH challenge.  Presumably this was avoided to minimize stressful handling, but this was not discussed.  Other potential confounds such as uneven sampling, unknown time of defecation within the sampling interval, and using wet feces vs. lyophilized feces--considering some were reported to be "a bit loose" (line 307) and thus may have had higher water content--could be explored further in the discussion.

Minor comments:

At times (for example line 13) the authors refer to FCMs as a method or tool, but it is the measurement of FCMs that is the tool.

The methods describe in cage behavioral tests, but only vaguely, and it is not clear what the purpose of this habituation was, when the animals could have been sham handled instead.  Referencing a source that describes these would be sufficient.

When referring to the place, "Arctic" is capitalized.

Some non-standard use of "e.g." that disrupt the flow of writing (lines 384 and 390 for example).

Lines 376-378--not clear why FCM eliminates the need for a separate control group.  This is true for repeated measures, but it would depend on the experiment.

Round 2

Reviewer 1 Report

Thank you for explaining the Larm et al. paper was performed by a separate research group in a different facility, that was not clear to me in the initial submission, and is sufficient reason for these to be separate manuscripts. Your studies are still very similar, but there is honestly insufficient replication of data in the literature so I don’t think that’s necessarily a drawback; it’s good to have multiple similar studies with the same basic findings. The information now included in the Introduction outlining some of the physical differences between white and blue color morphs is also helpful in distinguishing the two studies. Additionally, the Introduction in general flows much more smoothly now, and does a better job of presenting your research question and background information to the reader. I also appreciate the emphasis placed on reduction of stress from handling in these animals, evident throughout the manuscript and study design.